# Beyond Bandit Feedback
# in Online Multiclass Classification

**Dirk van der Hoeven**
dirk@dirkvanderhoeven.com
Dept. of Computer Science
Università degli Studi di Milano, Italy

**Federico Fusco**
fuscof@diag.uniroma1.it
Dept. of Computer, Control
and Management Engineering
Sapienza Università di Roma, Italy

**Nicolò Cesa-Bianchi**
nicolo.cesa-bianchi@unimi.it
DSRC & Dept. of Computer Science
Università degli Studi di Milano, Italy

## Abstract

We study the problem of online multiclass classification in a setting where the learner's feedback is determined by an arbitrary directed graph. While including bandit feedback as a special case, feedback graphs allow a much richer set of applications, including filtering and label efficient classification. We introduce GAP-PLETRON, the first online multiclass algorithm that works with arbitrary feedback graphs. For this new algorithm, we prove surrogate regret bounds that hold, both in expectation and with high probability, for a large class of surrogate losses. Our bounds are of order $B\sqrt{\rho K T}$, where $B$ is the diameter of the prediction space, $K$ is the number of classes, $T$ is the time horizon, and $\rho$ is the domination number (a graph-theoretic parameter affecting the amount of exploration). In the full information case, we show that GAPPLETRON achieves a constant surrogate regret of order $B^2 K$. We also prove a general lower bound of order $\max\left\{B^2 K, \sqrt{T}\right\}$ showing that our upper bounds are not significantly improvable. Experiments on synthetic data show that for various feedback graphs our algorithm is competitive against known baselines.

## 1 Introduction

In online multiclass classification a learner interacts with an unknown environment in a sequence of rounds. At each round $t$, the learner observes a feature vector $\boldsymbol{x}_t \in \mathbb{R}^d$ and outputs a prediction $y'_t$ for the label $y_t \in \{1, \ldots, K\}$ associated with $\boldsymbol{x}_t$. If $y'_t \neq y_t$, then the learner is charged with a mistake. Kakade et al. (2008) introduced the bandit version of online multiclass classification, where the only feedback received by the learner after each prediction is the loss $\mathbb{1}[y'_t \neq y_t]$. Hence, if a mistake is made at time $t$ (and $K > 2$), the learner cannot uniquely identify the true label $y_t$ based on the feedback information.

Although bandits are a canonical example of partial feedback, they fail to capture a number of important practical scenarios of online classification. Consider for example spam filtering, where an online learner is to classify emails as spam or non-spam based on their content. Whenever the learner classifies an email as legitimate, the recipient gets to see it, and can inform the learner whether the email was correctly classified of not. However, when the email is classified as spam, the learner does not get any feedback because the email is not checked by the recipient. Another example is label efficient multiclass classification. Here, instead of making a prediction, the learner can ask a human

35th Conference on Neural Information Processing Systems (NeurIPS 2021).

expert for the true label. At the steps when predictions are made, however, the learner does not receive any feedback information (not even their own loss). A further example is disease prevention: if we predict an outburst of disease in a certain area, we can preemptively stop it by vaccinating the local population. This intervention would prevent us from observing whether our prediction was correct, but would still allow us to observe an outburst occurring in a different area.

Unlike bandits, the amount of feedback obtained by the learner in these examples depends on the predicted class, and can vary from full information to no feedback at all. This scenario has been previously considered in the framework of online learning with feedback graphs (Mannor and Shamir, 2011; Alon et al., 2015, 2017). A feedback graph is a directed graph $\mathcal{G} = (\mathcal{V}, \mathcal{E})$ where each node in $\mathcal{V}$ receives at least one edge from some other node in $\mathcal{V}$ (possibly from itself). The nodes in $\mathcal{V}$ correspond to actions, and a directed edge $(a, b) \in \mathcal{E}$ indicates that by playing action $a$ the learner observes the loss of action $b$. This generalizes the well-known online learning settings of experts (where $\mathcal{G}$ is the complete graph, including self-loops) and bandits (where $\mathcal{G}$ has only self-loops). Note that it is easy to model spam filtering and label efficient prediction using feedback graphs. For spam filtering, $\mathcal{G}$ contains only two actions $s$ and $n$ (corresponding, respectively, to the learner's predictions for spam and non-spam), and the edge set is $\mathcal{E} = \big\{ (n, n), (n, s) \big\}$. For label efficient multiclass prediction, $\mathcal{G}$ contains a node for each class, plus an extra node corresponding to issuing a label request. It is important to observe that all previous analyses of feedback graphs only apply to the abstract setting of prediction with experts, where any dependence of the loss on feature vectors is ignored. This hampers the application of those results to online multiclass classification. In this work we build on previous results on online learning and classification with bandit feedback to design and analyze the first algorithm for online multiclass classification with arbitrary feedback graphs. In doing so, we also improve the analyses of the previously studied special cases (full information and bandit feedback) of multiclass classification.

In the online multiclass classification setting, the goal is bound the number of mistakes made by the learner. The mistake bounds take the following form:

$$\sum_{t=1}^{T} \mathbb{1}[y_t' \neq y_t] = \sum_{t=1}^{T} \ell_t(\boldsymbol{U}) + \mathcal{R}_T, \tag{1}$$

where $\ell_t$ is a surrogate loss, $\boldsymbol{U} \in \mathcal{W} \subseteq \mathbb{R}^{d \times K}$ is the matrix of reference predictors, and $\mathcal{R}_T$ is called the surrogate regret. In this work we provide two types of bounds on the surrogate regret: bounds that hold in expectation and bounds that hold with high probability. Note that equation (1) could also be written as $\sum_{t=1}^{T} \big( \mathbb{1}[y_t' \neq y_t] - \ell_t(\boldsymbol{U}) \big) = \mathcal{R}_T$. However, we prefer the former former since $\mathcal{R}_T$ is not a proper regret: because the zero-one loss is non-convex we compare it with a surrogate loss.

Our results build on recent work by Van der Hoeven (2020), who showed that one can exploit the gap between the surrogate loss and the zero-one loss to derive improved surrogate regret bounds in the full information and bandit settings of online multiclass classification. We modify the GAP-TRON algorithm (Van der Hoeven, 2020) to make it applicable to the feedback graph setting. In the analysis of the resulting algorithm, called GAPPLETRON[1], we use several new insights to show that it has $O(B\sqrt{\rho K T})$ surrogate regret in expectation and $O\big(\sqrt{\rho K T (B^2 + \ln(1/\delta))}\big)$ surrogate regret with probability at least $1 - \delta$ for any feedback graph with domination number[2] $\rho$, and for any $\big\|\mathrm{vec}(\boldsymbol{U})\big\| \leq B$ for some norm $\|\cdot\|$ (if $\|\cdot\|$ is the Euclidean norm, then $\big\|\mathrm{vec}(\boldsymbol{U})\big\|$ is the Frobenius norm of $\boldsymbol{U}$). For example, in both spam filtering and label efficient classification we have $\rho = 1$. So in the label efficient setting, where each label request counts as a mistake, with high probability GAPPLETRON makes at most order of $B\sqrt{KT}$ mistakes while requesting at most order of $B\sqrt{KT}$ labels. Note that we are not aware of previously known high-probability bounds on the surrogate regret. Furthermore, whereas the results of Van der Hoeven (2020) only hold for a limited number of surrogate loss functions, our results hold for the larger class of regular surrogate losses.

Interestingly, with feedback graphs the surrogate regret for online multiclass classification has, in general, a better dependence on $T$ than the regret for online learning. Indeed, Alon et al. (2015) show that the best possible online learning regret is $\Omega(T^{2/3})$ for certain feedback graphs that are

---

[1]Our algorithm is called after the apple tasting feedback model, which is the original name of the spam filtering graph.

[2]The domination number is the cardinality of the smallest dominating set.

| Upper bounds | Partial | Full |
|---|---|---|
| Non-separable | $B\sqrt{\rho KT}$ | $KB^2$ |
| Separable | $B\sqrt{\rho T}$ | $B^2$ |
| **Lower bounds** | | |
| Non-separable | | $KB^2$ |
| Separable | $\sqrt{T}$ | $B^2$ |

Table 1: Overview of the surrogate regret bounds in the separable and non-separable case. The upper bounds hold with high probability, while the lower bounds apply to any randomized prediction algorithm. All bounds are novel except for the lower bound in the full information separable case (Beygelzimer et al., 2019, Theorem 11).

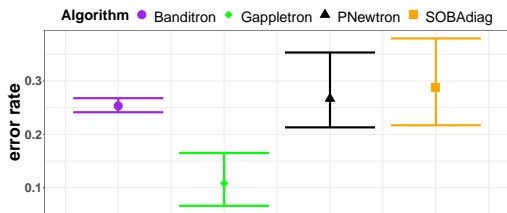

Figure 1: Error rate in non-separable synthetic bandit experiments showcasing GAPPLETRON against known baselines. The points are the means and the whiskers are minimum and maximum error rate over ten repetitions (details in Section 6).

called weakly observable (e.g., the graphs for label efficient classification). In contrast, we prove a $O(T^{1/2})$ upper bound on the surrogate regret for any feedback graph, including weakly observable ones.

Our results cannot be significantly improved in general: we prove a $\Omega(B^2 K + \sqrt{T})$ lower bound on the surrogate regret. Due to the new insights required by their proofs, we believe the high-probability bounds and the lower bounds are our strongest technical contributions.

We provide several other new results. In the separable case, when there exists a $U$ for which $\sum_{t=1}^{T} \ell_t(U) = 0$, GAPPLETRON has $O(B\sqrt{\rho T})$ surrogate regret in expectation. Even though $O(B^2 K)$ mistake bounds are possible in the separable setting (Beygelzimer et al., 2019), ours is the first algorithm that has satisfactory surrogate regret in the non-separable case and has an improved surrogate regret in the separable case. Note that although BANDITRON (Kakade et al., 2008) also makes $O(B\sqrt{KT})$ mistakes in the separable case, it suffers $O(K^{1/3}(BT)^{2/3})$ surrogate regret in the non-separable case. Our results for the separable case in the full information setting improve results of Van der Hoeven (2020) by a factor of $K$: GAPPLETRON suffers $O(B^2)$ surrogate regret both in expectation and with high probability, thus matching the bounds of the classical PERCEPTRON algorithm (Rosenblatt, 1958; Novikov, 1962) — see Table 1 for a summary of our theoretical results. Finally, we also evaluated the performance of GAPPLETRON in several experiments, showing that GAPPLETRON is competitive against known baselines in the full information, bandit, and multiclass spam filtering setting, in which predicting a certain class provides full information feedback and all other predictions do not provide any information (see Figure 1 for an experimental result in the bandit setting).

**Additional related work.** The full information and bandit versions of the online multiclass classification setting have been extensively studied. Here we provide the most relevant references and defer the reader to Van der Hoeven (2020) for a more extensive literature review. Algorithms for the full information setting include: the PERCEPTRON, its multiclass versions (Rosenblatt, 1958; Crammer and Singer, 2003; Fink et al., 2006) and many variations thereof, second-order algorithms such as AROW (Crammer et al., 2009) and the second-order PERCEPTRON (Cesa-Bianchi et al., 2005), and various algorithms for online logistic regression — see Foster et al. (2018) and references therein. In the bandit setting, we mention the algorithms BANDITRON (Kakade et al., 2008), NEWTRON (Hazan and Kale, 2011), SOBA (Beygelzimer et al., 2017), and OBAMA (Foster et al., 2018).

Online learning with feedback graphs has been investigated both in the adversarial and stochastic regimes. In the adversarial setting, variants where the graph changes over time and is partially known or stochastic have been studied by Cohen et al. (2016); Kocák et al. (2016). Regret bounds that scale with the loss of the best action have been obtained by Lykouris et al. (2018). Other variants include sleeping experts (Cortes et al., 2019), switching experts (Arora et al., 2019), and adaptive adversaries (Feng and Loh, 2018). Some works use feedback graphs to bound the regret in auctions (Cesa-Bianchi et al., 2017; Feng et al., 2018; Han et al., 2020). In the stochastic setting, regret

bounds for Thompson sampling and UCB have been analyzed by Tossou et al. (2017); Liu et al. (2018); Lykouris et al. (2020). Finally, feedbacks graphs can also be viewed as a special case of the partial monitoring framework for sequential decisions, see (Lattimore and Szepesvári, 2020) for an introduction to the area.

Helmbold et al. (2000) introduced online filtering as "apple tasting". However, their analysis applies to a restricted version of online learning in which instances $x_t$ belong to a finite domain, and the labels $y_t$ are such that $y_t = f(x_t)$ for all $t$ and for some fixed $f$ in a known class of functions. Practical applications of online learning to spam filtering have been investigated by Cesa-Bianchi et al. (2003); Sculley (2008).

**Notation.**   Let $\mathbf{1}$ and $\mathbf{0}$ denote, respectively, the all-one and all-zero vectors, and let $\boldsymbol{e}_k$ be the basis vector in direction $k$. Let $[K] = \{1, \ldots, K\}$ and let $\mathbb{R}_+$ be the non-negative real numbers. We use $\langle \boldsymbol{g}, \boldsymbol{w} \rangle$ to denote the inner product between vectors $\boldsymbol{g}, \boldsymbol{w} \in \mathbb{R}^d$. The rows of matrix $\boldsymbol{W} \in \mathbb{R}^{K \times d}$ are denoted by $\boldsymbol{W}^1, \ldots, \boldsymbol{W}^K$. Whenever possible, we use the same symbol $\boldsymbol{W}$ to denote both a $K \times d$ matrix and a column vector $\mathrm{vec}(\boldsymbol{W}) = (\boldsymbol{W}^1, \ldots, \boldsymbol{W}^K)$ in $\mathbb{R}^{Kd}$. We use $\|\boldsymbol{x}\|_2$ to denote the Euclidean norm of a vector $\boldsymbol{x}$ and $\|\boldsymbol{x}\|$ to denote an arbitrary norm. The Kronecker product between matrices $\boldsymbol{W}$ and $\boldsymbol{U}$ is denoted by $\boldsymbol{W} \otimes \boldsymbol{U}$. We assume $\boldsymbol{W} \in \mathcal{W}$ for some convex $\mathcal{W} \subseteq \mathbb{R}^{K \times d}$. This is equivalent to say that $\mathrm{vec}(\boldsymbol{W})$ belongs to a convex subset of $\mathbb{R}^{Kd}$, for example a $p$-norm ball. As in previous works, we assume instance-label pairs $(\boldsymbol{x}_t, y_t)$ are generated by an adversary who is oblivious to the algorithm's internal randomization. Finally, for any round $t$, $P_t[\cdot]$ and $\mathbb{E}_t[\cdot]$ denote the conditional probability and expectation, given the randomized predictions $y_1', y_2', \ldots, y_{t-1}'$ and the corresponding feedback.

A feedback graph is any directed graph $\mathcal{G} = (\mathcal{V}, \mathcal{E})$, with edges $\mathcal{E}$ and nodes $\mathcal{V}$, such that for any $y \in \mathcal{V}$ there exists some $y' \in \mathcal{V}$ such that $(y', y) \in \mathcal{E}$, where we allow $y' = y$. In online multiclass classification, $\mathcal{V} = [K]$ and $\mathcal{E}$ specifies which predictions observe which outcomes. Let $\mathrm{out}(y') = \{y \in \mathcal{V} : (y', y) \in \mathcal{E}\}$ be the out-neighbourhood of $y'$. If the learner predicts $y_t'$ at time $t$, then the feedback received by the learner is the set of pairs $\big(y, \mathbb{1}[y \neq y_t]\big)$ for all $y \in \mathrm{out}(y')$. Due to the structure of the zero-one loss, if a node has $K - 1$ outgoing edges, we always add the missing edge to $\mathcal{E}$ as this does not change the information available to the learner. We say that an outcome $y'$ is revealing if predicting that outcome provides the learner with full information feedback, i.e., $\mathrm{out}(y') = [K]$, and we denote the set of revealing outcomes by $\mathcal{Q}$. For example, in label efficient classification, querying the true label $y_t$ corresponds to playing a revealing outcome. We say that a set of nodes $\mathcal{S}$ is a dominating set if for each $y \in \mathcal{V}$ there is a node $y' \in \mathcal{S}$ such that $y \in \mathrm{out}(y')$. The number of nodes in a smallest dominating set is called the domination number, and we denote it by $\rho$. Note that GAPPLETRON is run using the minimum dominating set $\mathcal{S}$, which is known to be hard to recover in general. However, if the algorithm is fed with any other dominating set $\mathcal{S}'$ of bigger cardinality $\rho'$, our results continue to hold with $\rho$ replaced by $\rho'$ (recall that a dominating set of size at most $(\ln \rho + 2)\rho$ can be efficiently found via a greedy approximation algorithm).

**Regular surrogate losses.**   Fix a convex domain $\mathcal{W}$. Let $\ell : \mathcal{W} \times \mathbb{R}^d \times [K] \to \mathbb{R}_+$ be any function convex on $\mathcal{W}$ such that, for all $\boldsymbol{W} \in \mathcal{W}$, $\boldsymbol{x} \in \mathbb{R}^d$, and $y \in [K] \setminus \{y^\star\}$ (with $y^\star = \arg\max_k \langle \boldsymbol{W}^k, \boldsymbol{x} \rangle$) we have

$$\frac{K-1}{K} \ell(\boldsymbol{W}, \boldsymbol{x}, y) + \frac{1}{K} \ell(\boldsymbol{W}, \boldsymbol{x}, y^\star) \geq 1. \tag{2}$$

Then $\ell_t = \ell(\cdot, \boldsymbol{x}_t, y_t)$ is a regular surrogate loss if

$$\|\nabla \ell_t(\boldsymbol{W})\|^2 \leq 2L\, \ell_t(\boldsymbol{W}) \qquad \boldsymbol{W} \in \mathcal{W} \tag{3}$$

for some norm $\|\cdot\|$. When $\|\cdot\|$ is the Euclidean norm, the condition on the gradient is satisfied by all $L$-smooth surrogate loss functions (see, for example, (Zhou, 2018, Lemma 4)).

Examples of regular surrogate losses are the smooth hinge loss (Rennie and Srebro, 2005) and the logistic loss with base $K$, defined by $\ell_t(\boldsymbol{W}_t) = -\log_K q(\boldsymbol{W}_t, \boldsymbol{x}_t, y_t)$, where $q$ is the softmax function. Even though the hinge loss is not a regular surrogate loss, in Appendix A we show that a particular version of the hinge loss satisfies all the relevant properties of regular surrogate losses. Also, note that in the feedback graph setting, this particular version of the hinge loss we use is random whenever the learner's predictions are randomized.

## 2 Gappletron

---

Algorithm 1: GAPPLETRON

---

**Input:** Set of revealing actions $\mathcal{Q} \subseteq [K]$, minimum dominating set $\mathcal{S}$, OCO algorithm $\mathcal{A}$ with
domain $\mathcal{W} \subseteq \mathbb{R}^d$, $\gamma \geq 0$, and gap map $a : \mathbb{R}^{K \times d} \times \mathbb{R}^d \to [0, 1]$

1: **for** $t = 1 \ldots T$ **do**
2:      Obtain $\boldsymbol{x}_t$
3:      Let $y_t^\star = \arg\max_k \langle \boldsymbol{W}_t^k, \boldsymbol{x}_t \rangle$                 ▷ max-margin prediction
4:      **if** $y_t^\star \in \mathcal{Q}$ **then**
5:          Set $\gamma_t = 0$
6:      **else**
7:          Set $\gamma_t = \min\left\{\frac{1}{2}, \gamma \big/ \sqrt{\left|\{s \leq t : y_s^\star \notin \mathcal{Q}\}\right|}\right\}$          ▷ exploration rate
8:      Set $\zeta_t = \mathbb{1}[\gamma_t \leq a(\boldsymbol{W}_t, \boldsymbol{x}_t)]$
9:      $\boldsymbol{p}_t' = \left(1 - \zeta_t a(\boldsymbol{W}_t, \boldsymbol{x}_t) - (1 - \zeta_t)\gamma_t\right)\boldsymbol{e}_{y_t^\star} + \zeta_t a(\boldsymbol{W}_t, \boldsymbol{x}_t)\frac{1}{K}\mathbf{1} + (1 - \zeta_t)\frac{\gamma_t}{\rho}\mathbf{1}_S$
10:      Predict with label $y_t' \sim \boldsymbol{p}_t'$
11:      Compute $v_t = \dfrac{\mathbb{1}[y_t \in \text{out}(y_t')]}{P_t(y_t \in \text{out}(y_t'))}$          ▷ $y_t$ is observed only when $y_t \in \text{out}(y_t')$
12:      Set $\widehat{\ell}_t(\boldsymbol{W}_t) = v_t\ell_t(\boldsymbol{W}_t)$          ▷ compute loss estimates
13:      Send $\widehat{\ell}_t$ to $\mathcal{A}$ and get $\boldsymbol{W}_{t+1}$ in return

---

In this section we introduce GAPPLETRON, whose pseudocode is presented in Algorithm 1. As input,
the algorithm takes information about the graph $\mathcal{G}$ in the form of a minimum dominating set $\mathcal{S}$ and
a (possibly empty) set of revealing actions $\mathcal{Q}$. GAPPLETRON maintains a parameter $\boldsymbol{W}_t \in \mathcal{W} \subseteq \mathbb{R}^{d \times K}$ and uses some full information Online Convex Optimization (OCO) algorithm $\mathcal{A}$ to update
the vector form of $\boldsymbol{W}_t$. Our results hold whenever $\mathcal{A}$ satisfies the condition that $\sum_{t=1}^T \left(\widehat{\ell}_t(\boldsymbol{W}_t) - \widehat{\ell}_t(\boldsymbol{U})\right)$ be at most of order $h(\boldsymbol{U})\sqrt{\sum_{t=1}^T \|\nabla\widehat{\ell}_t(\boldsymbol{W}_t)\|^2}$, where $\widehat{\ell}_t$ are the estimated losses computed
at line 12 of Algorithm 1 and $h : \mathcal{W} \to \mathbb{R}_+$ is any upper bound on the norm of $\boldsymbol{U} \in \mathcal{W}$. Since
practically any OCO algorithm can be tuned to have such a guarantee — see (Orabona and Pál,
2018) — this is a mild requirement. Whereas GAPTRON is only able to use Online Gradient Descent
(OGD) with a fixed learning rate, GAPPLETRON allows for more flexibility, which in turn may lead
to different guarantees on the surrogate regret. For example, if the learner runs an OCO algorithm
with a good dynamic regret bound (Zinkevich, 2003), then GAPPLETRON enjoys a good dynamic
surrogate regret bound. Furthermore, the guarantee of $\mathcal{A}$ allows us to derive stronger results in the
separable setting while maintaining a similar guarantee as GAPTRON in the non-separable setting,
which is not possible when using OGD with a fixed learning rate. Additional inputs to GAPPLETRON
are $\gamma > 0$, which is used to control the exploration rate of the algorithm in the partial information
setting, and the gap map $a$, whose role we explain below.

The predictions of Algorithm 1 are sampled from $\boldsymbol{p}_t'$ defined in line 9, where $\boldsymbol{e}_{y_t^\star}$ is the basis vector
in the direction of the margin-based linear prediction $y_t^\star = \arg\max_k \langle \boldsymbol{W}_t^k, \boldsymbol{x}_t \rangle$. The gap map
$a : \mathbb{R}^{K \times d} \times \mathbb{R}^d \to [0, 1]$ controls the mixture between $\boldsymbol{e}_{y_t^\star}$ and the uniform exploration term $\frac{1}{K}\mathbf{1}$.
For brevity, we sometimes write $a_t$ instead of $a(\boldsymbol{W}_t, \boldsymbol{x}_t)$. The single most important property of
GAPPLETRON is presented in the following Lemma.

**Lemma 1.** *Fix any feedback graph $\mathcal{G}$ and suppose that, for all $t$, $\ell_t$ is a regular surrogate loss with
respect to $\ell$. Then GAPPLETRON, run on $\mathcal{G}$ with $a$ such that $a(\boldsymbol{W}_t, \boldsymbol{x}_t) = \ell(\boldsymbol{W}_t, \boldsymbol{x}_t, y_t^\star)$, satisfies*

$$\sum_{y \in [K]} p_t'(y)\mathbb{1}[y \neq y_t] \leq \frac{K-1}{K}\ell_t(\boldsymbol{W}_t) + \gamma_t.$$

*Proof.* First, observe that $\sum_{y \in [K]} p_t'(y)\mathbb{1}[y \neq y_t] \leq (1 - a_t)\mathbb{1}[y_t^\star \neq y_t] + a_t\frac{K-1}{K} + \gamma_t$, since $\zeta$, $(1 - \zeta)$, and the cost of exploration are at most 1. To conclude the proof we claim that the first two terms
in the right-hand side are upper bounded by $\frac{K-1}{K}\ell_t(\boldsymbol{W}_t)$. We show that by considering two cases.
In the first case $y_t^\star = y_t$ and the inequality simply follows by substituting $a_t = \ell(\boldsymbol{W}_t, \boldsymbol{x}_t, y_t^\star) = $

$\ell_t(\boldsymbol{W}_t)$. In the second case $y_t^\star \neq y_t$ and we have that

$$(1 - a_t)\mathbb{1}[y_t^\star \neq y_t] + a_t \frac{K-1}{K} = 1 - \frac{1}{K}\ell(\boldsymbol{W}_t, \boldsymbol{x}_t, y_t^\star)$$

$$= 1 - \frac{1}{K}\ell(\boldsymbol{W}_t, \boldsymbol{x}_t, y_t^\star) - \frac{K-1}{K}\ell(\boldsymbol{W}_t, \boldsymbol{x}_t, y_t) + \frac{K-1}{K}\ell_t(\boldsymbol{W}_t) \leq \frac{K-1}{K}\ell_t(\boldsymbol{W}_t),$$

where the inequality is due to equation (2) in the definition of regular surrogate losses. $\qquad\square$

Although the GAPTRON algorithm uses similar predictions, it is not clear how to choose $a$ such that a property similar to the one described in Lemma 1 holds. Rather, Van der Hoeven (2020) derives a different gap map for the hinge loss, the smooth hinge loss, and the logistic loss, and analyses the surrogate regret separately for each loss. With Lemma 1 in hand, we simplify the analysis and — at the same time — also generalize the results of Van der Hoeven (2020) to other surrogate losses. Furthermore, Lemma 1 also allows us derive surrogate regret bounds that hold with high probability.

What Lemma 1 states is that with regular surrogate losses and $a(\boldsymbol{W}_t, \boldsymbol{x}_t) = \ell(\boldsymbol{W}_t, \boldsymbol{x}_t, y_t^\star)$ the expected zero-one loss of GAPPLETRON can be upper bounded by $\frac{K-1}{K}\ell_t(\boldsymbol{W}_t)$ plus the cost of exploration. While at first this may seem of little interest, note that we want to bound the zero-one loss in terms of $\ell_t$ rather than $\frac{K-1}{K}\ell_t$. Compared to standard algorithms, this gains us a $-\frac{1}{K}\ell_t(\boldsymbol{W}_t)$ term in *each* round, which we can use to derive our results. To see how, observe that GAPPLETRON uses an OCO algorithm $\mathcal{A}$ to update $\mathrm{vec}(\boldsymbol{W}_t)$ on each round. Suppose that, for some $h : \mathcal{W} \to \mathbb{R}$ and $\boldsymbol{U} \in \mathcal{W}$, Algorithm $\mathcal{A}$ satisfies

$$\sum_{t=1}^{T}\left(\widehat{\ell}_t(\boldsymbol{W}_t) - \widehat{\ell}_t(\boldsymbol{U})\right) \leq h(\boldsymbol{U})\sqrt{\sum_{t=1}^{T}\|\widehat{\boldsymbol{g}}_t\|^2}, \tag{4}$$

where $\widehat{\boldsymbol{g}}_t = v_t \nabla \ell_t(\boldsymbol{W}_t)$. For simplicity, now assume we are in the full information setting (e.g., $v_t = 1$ for all $t$). Since $\ell_t$ is a regular surrogate loss, we can use $\|\nabla \ell_t(\boldsymbol{W})\|^2 \leq 2L\,\ell_t(\boldsymbol{W})$ and $\sqrt{ab} = \frac{1}{2}\inf_{\eta>0}\{a/\eta + \eta b\}$ to show that

$$h(\boldsymbol{U})\sqrt{\sum_{t=1}^{T}\|\widehat{\boldsymbol{g}}_t\|^2} - \sum_{t=1}^{T}\frac{1}{K}\ell_t(\boldsymbol{W}_t) \leq h(\boldsymbol{U})\sqrt{\sum_{t=1}^{T}2L\ell_t(\boldsymbol{W}_t)} - \sum_{t=1}^{T}\frac{1}{K}\ell_t(\boldsymbol{W}_t) \leq \frac{KLh(\boldsymbol{U})^2}{2}.$$

This means that in the full information setting the surrogate regret of GAPPLETRON is independent of the number of rounds. In the partial information setting some additional steps are required, but the idea remains essentially the same. We formalize the aforementioned ideas in the following Lemma, whose proof is deferred to Appendix B.

**Lemma 2.** *Fix any feedback graph $\mathcal{G}$ and suppose that, for all $t$, $\ell_t$ is a regular surrogate loss with respect to $\ell$. If $\mathcal{A}$ satisfies equation (4) then, for any realization of the randomized predictions $y_1', \ldots, y_T'$, GAPPLETRON, run on $\mathcal{G}$ with gap map $a$ such that $a(\boldsymbol{W}_t, \boldsymbol{x}_t) = \ell(\boldsymbol{W}_t, \boldsymbol{x}_t, y_t^\star)$, satisfies*

$$\sum_{t=1}^{T}\sum_{y \in [K]} p_t'(y)\mathbb{1}[y \neq y_t] \leq \sum_{t=1}^{T}\widehat{\ell}_t(\boldsymbol{U}) + \sum_{t=1}^{T}\gamma_t$$

$$+ \inf_{\eta>0}\left\{\frac{h(\boldsymbol{U})^2}{2\eta} + \sum_{t=1}^{T}\left(\frac{K-1}{K}\ell_t(\boldsymbol{W}_t) - v_t\ell_t(\boldsymbol{W}_t) + \eta v_t^2 L\ell_t(\boldsymbol{W}_t)\right)\right\} \qquad \forall \boldsymbol{U} \in \mathcal{W}\,.$$

## 3 Bounds that Hold in Expectation

In this section we present bounds on the surrogate regret that hold in expectation. For brevity we use $\mathcal{M}_T = \sum_{t=1}^{T}\mathbb{1}[y_t' \neq y_t]$. We now state a simplified version of Theorem 4, whose full statement and proof can be found in Appendix C.

**Theorem 1.** *Let $\mathcal{G}$ be any feedback graph with dominating number $\rho$ and revealing action set $\mathcal{Q}$. Suppose that, for all $t$, $\ell_t$ is a regular surrogate loss with respect to $\ell$. If $\mathcal{A}$ satisfies equation (4) then GAPPLETRON, run on $\mathcal{G}$ and $\mathcal{A}$ with gap map $a$ such that $a(\boldsymbol{W}_t, \boldsymbol{x}_t) = \ell(\boldsymbol{W}_t, \boldsymbol{x}_t, y_t^\star)$, satisfies*

$$\mathbb{E}[\mathcal{R}_T] = O\left(\mathbb{E}\left[\max\left\{\frac{K^2 Lh(\boldsymbol{U})^2}{\max\{1, |\mathcal{Q}|\}}, h(\boldsymbol{U})\sqrt{\rho K L|\{t : y_t^\star \notin \mathcal{Q}\}|}\right\}\right]\right) \qquad \forall \boldsymbol{U} \in \mathcal{W}\,.$$

*Furthermore, for all $\boldsymbol{U} \in \mathcal{W}$ such that $\sum_{t=1}^T \ell_t(\boldsymbol{U}) = 0$,* GAPPLETRON *satisfies:*

$$\mathbb{E}\left[\mathcal{M}_T\right] = O\left(\mathbb{E}\left[\max\left\{h(\boldsymbol{U})\sqrt{\rho L|\{t : y_t^\star \notin \mathcal{Q}\}|}, \frac{KLh(\boldsymbol{U})^2}{\max\{1, |\mathcal{Q}|\}}\right\}\right] - \frac{1}{K}\mathbb{E}\left[\sum_{t=1}^T \ell_t(\boldsymbol{W}_t)\right]\right).$$

In the full information setting we clearly have that $\mathcal{Q} = [K]$. Hence, using OGD as $\mathcal{A}$ with an appropriated learning rate, the second statement in Theorem 1 reduces to $\mathbb{E}\left[\mathcal{M}_T\right] \leq 4L\|\boldsymbol{U}\|_2^2 - \sum_{t=1}^T \frac{1}{K}\ell_t(\boldsymbol{W}_t)$, which improves the results of GAPTRON in the separable case by at least a factor $K$. Interestingly, compared to standard bounds for the separable case, such as the PERCEPTRON bound, there is a negative term which seems to further lower the cost of learning how to separate the data. Similarly, in the partial information setting, the bound for the separable case in Theorem 1 has a reduced dependency on $K$ compared to the non-separable case, obtaining similar improvements over GAPTRON as in the full information setting.

For the non-separable case, Theorem 1 generalizes GAPTRON in two directions. The most prominent direction is the extension is to feedback graphs, where our analysis reveals a surprising phenomenon: Theorem 1 in fact shows that the surrogate regret in the label efficient setting (and in any setting where $\rho < K$) is actually smaller than in the bandit setting, where $\rho = K$. Intuitively, this is due to the fact that our algorithm only updates when $y_t$ is known. In the bandit setting, we need to explore all labels to find $y_t$, while in label efficient classification we can just play whichever action is the revealing action, and find $y_t$. This implies that exploration in label efficient classification is easier than in the bandit setting. Note that in the bandit setting, playing $y_t' \neq y_t$ also provides the learner with information. Perhaps by using this information effectively, one is able to improve our surrogate regret bounds, but as of yet it is not clear how to use knowledge of the wrong label. The second extension is that the bounds in Theorem 1 hold for all regular surrogate loss functions with the same gap map defined by the surrogate loss, rather than only for a limited number of loss functions and ad-hoc gap maps as it was the case with GAPTRON.

## 4 Bounds that Hold with High Probability

We now present bounds on the surrogate regret that hold with high probability. After proving a general surrogate regret bound, we derive a corresponding bound, with improved guarantees, for the full information setting. The bound for the partial information setting can be found in Theorem 5 in Appendix D, which implies Theorem 2 below. Let the maximum loss over all rounds be $\ell_{\max} = \max_{t, \boldsymbol{W} \in \mathcal{W}} \ell_t(\boldsymbol{W})$.

**Theorem 2.** *With probability at least $1 - \delta$,* GAPPLETRON *satisfies:*

$$\mathcal{R}_T = O\left(\sqrt{(Lh(\boldsymbol{U})^2 + \ell_{\max}\ln(1/\delta))K\rho T}\right) \qquad \forall \boldsymbol{U} \in \mathcal{W}$$

*Furthermore, for all $\boldsymbol{U} \in \mathcal{W}$ such that $\sum_{t=1}^T \ell_t(\boldsymbol{U}) = 0$, with probability at least $1 - \delta$* GAP-PLETRON *satisfies:*

$$\mathcal{M}_T = O\left(\sqrt{(Lh(\boldsymbol{U})^2 + K\ell_{\max}\ln(1/\delta))\rho T}\right).$$

Theorem 2 shows that Algorithm 1 has $O(h(\boldsymbol{U})\sqrt{\rho K T})$ surrogate regret in the worst case, with high probability. As far as the authors are aware, this is the first high-probability surrogate regret bound for a margin-based classifier in the partial information setting. Similarly to the bounds in expectation, the worst-case surrogate regret is the largest in the bandit setting ($\rho = K$) and the smallest in label efficient classification ($\rho = 1$). Unlike the bounds in expectation, where the surrogate regret was at least a factor $\sqrt{K}$ smaller in the separable case, the improvement in Theorem 2 is less apparent, but the surrogate regret still has a better dependence on $K$ in the separable case. In particular, all the terms with $h(\boldsymbol{U})$ have a better dependence on $K$.

In the full information setting the dependence on $\ell_{\max}$ can be removed. This cannot be achieved in the partial information setting, due to the necessity of estimating the surrogate loss. If $\mathcal{W}$ has a bounded radius $B$ and $\ell_t$ has gradients bounded by $G$, then $\ell_{\max} \leq 1 + BG$ by convexity. The bound for the full information setting can be found in Theorem 3. In the separable case of the full information setting, the bound does not depend on $K$, which is not the case for Theorem 2 due to the need to control the surrogate loss estimates.

**Theorem 3.** *Under the conditions of Lemma 2, with probability at least* $1-\delta$, GAPPLETRON *satisfies*

$$\mathcal{M}_T \leq \sum_{t=1}^{T} \ell_t(\boldsymbol{U}) + KLh(\boldsymbol{U})^2 + \frac{3K \ln(1/\delta)}{2} \qquad \forall \boldsymbol{U} \in \mathcal{W} \ .$$

*Furthermore, for all* $\boldsymbol{U} \in \mathcal{W}$ *such that* $\sum_{t=1}^{T} \ell_t(\boldsymbol{U}) = 0$, *then with probability at least* $1 - \delta$, GAPPLETRON *satisfies* $\mathcal{M}_T \leq 4Lh(\boldsymbol{U})^2 + \frac{3}{4} \ln \frac{1}{\delta}$.

We provide a proof sketch of the full information versions of Theorem 3. The proof for the partial information setting is essentially the same, with some extra steps to control the estimates of the surrogate losses. Let $z_t = \left( \mathbb{1}[y_t' \neq y_t] - \sum_{y \in [K]} p_t'(y) \mathbb{1}[y \neq y_t] \right)$. The proof relies on (Beygelzimer et al., 2011, Theorem 1) — see Lemma 4 in this paper, which, when translated to our setting, states that with probability at least $1 - \delta$, for $\eta \in [0, 1]$,

$$\sum_{t=1}^{T} z_t \leq \frac{1}{\eta} \ln \frac{1}{\delta} + \eta \sum_{t=1}^{T} \mathbb{E}_t \left[ z_t^2 \right] \ .$$

Since the variance is bounded by the second moment, $\mathbb{E}_t \left[ z_t^2 \right] \leq \mathbb{E}_t \left[ \mathbb{1}[y_t' \neq y_t] \right] \leq \frac{K-1}{K} \ell_t(\boldsymbol{W}_t)$, where the last inequality is due to Lemma 1. By applying Lemma 2, we find that, for some $\eta \in [0, 1]$,

$$\mathcal{M}_T \leq \sum_{t=1}^{T} \ell_t(\boldsymbol{U}) + \frac{h(\boldsymbol{U}) - \ln \delta}{\eta} + \sum_{t=1}^{T} \left( \eta \left( L + \frac{K - 1}{K} \right) - \frac{1}{K} \right) \ell_t(\boldsymbol{W}_t),$$

with probability at least $1-\delta$. After choosing an appropriate $\eta$, this gives us a $O\left(Kh(\boldsymbol{U})^2\right)$ surrogate regret bound with high probability.

## 5 Lower Bounds

Corollary 1 below here shows that the bound of Theorem 1 cannot be significantly improved.

**Corollary 1.** *Let $A$ be a possibly randomized algorithm for the online multiclass classification setting with feedback graphs. Then, for any $B = \Omega(1)$, the surrogate regret of $A$ with respect to the smooth hinge loss must satisfy*

$$\mathbb{E}\left[\mathcal{M}_T\right] = \min_{\boldsymbol{U} \in \mathcal{W}} \sum_{t=1}^{T} \ell_t(\boldsymbol{U}) + \Omega\left(KB^2 + \sqrt{T}\right)$$

*where $K$ is the number of classes, the feature vectors $\boldsymbol{x}_t$ satisfy $\left\|\boldsymbol{x}_t\right\|_2 = \Theta(1)$ for all t, and $\mathcal{W} = \left\{\boldsymbol{W} : \left\|\boldsymbol{W}\right\| \leq B\right\}$.*

Corollary 1 is implied by Theorems 6 and 7 in Appendix E. The proof of Theorem 6 builds on the lower bound of Daniely et al. (2015) for strongly-adaptive regret. The feedback graph considered in the proof is filtering with two classes: a blind class (no outgoing edges) and a revealing class. In the proof, we show that the algorithm either explores too much, in which case the lower bound trivially holds, or the algorithm explores too little, in which case the environment can trick the algorithm into playing the wrong action by exploiting the blind class.

## 6 Experiments

We empirically evaluated the performance of GAPPLETRON on synthetic data in the bandit, multiclass filtering, and full information settings. Similarly to the SynSep and NonSynSyp datasets described in (Kakade et al., 2008), we generated synthetic datasets with $d \in \{80, 120, 160\}$, $K \in \{6, 9, 12\}$, and the label noise rate in $\{0, 0.05, 0.1\}$. Due to space constraints, we only report part of the experiments for the bandit setting in the main text, see Figure 2. In the bandit setting we used worst case tuning for the algorithms with the parameters suggested by theory, or set all parameters to 1, except for $T$. Initially we only used theoretical tuning for all algorithms, but we found that two algorithms we compared with did not have satisfactory results. A more detailed description

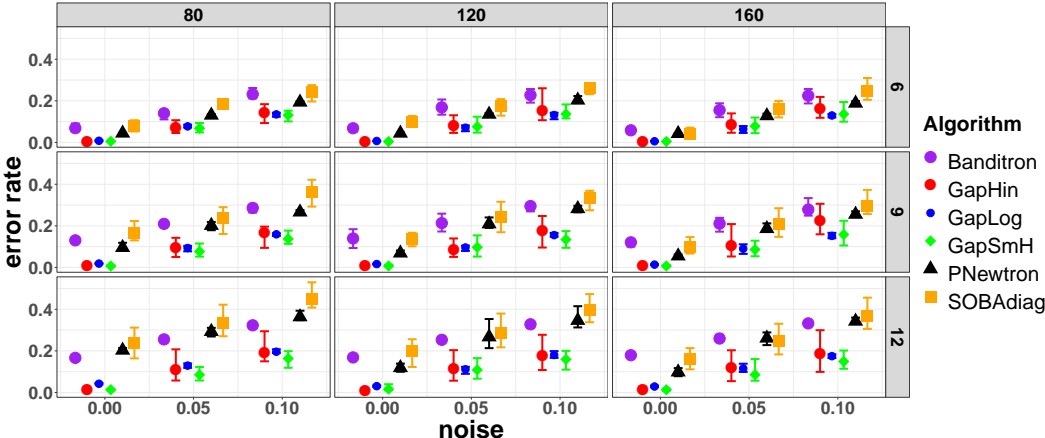

Figure 2: Results of the synthetic experiments for the bandit setting. The plot shows the best results of algorithms with parameters suggested by theory, or tuned with all parameters set to 1, except for $T$. The rows indicate different values for $K$ and the columns different values for $d$. Whiskers show the minimum and the maximum error rate over ten repetitions.

of the results, including how we generated data and tuned the algorithms, can be found in Appendix F.

In the bandit setting, we compared GAPPLETRON with the following baselines: PNewtron (the diagonal version of Newtron, by Hazan and Kale (2011)), SOBAdiag (the diagonal version of SOBA, by Beygelzimer et al. (2017)), and the importance weighted version of Banditron[3] (Kakade et al., 2008). We opted to use the diagonal versions of Newtron and SOBA for computational reasons. We chose the importance weighted version of Banditron because the standard version did not produce satisfactory results. We used three surrogate losses for GAPPLETRON: the logistic loss $\ell_t(\boldsymbol{W}_t) = -\log_K q(\boldsymbol{W}_t, \boldsymbol{x}_t, y_t)$ where $q$ is the softmax, the hinge loss defined in (5), and the smooth hinge loss (Rennie and Srebro, 2005), denoted by GapLog, GapHin, and GapSmH respectively. The OCO algorithm used with all losses is Online Gradient Descent, with learning rate $\eta_t = \left(10^{-8} + \sum_{j=1}^{t} \|\nabla \widehat{\ell}_j(\boldsymbol{W}_t)\|_2^2\right)^{-1/2}$ and no projections.

As shown in Figure 2, on average all versions of GAPPLETRON outperform the baselines in the bandit setting. GapHin appears to be more unstable than the other versions of GAPPLETRON. We suspect this is due to the fact that GapHin explores less than its counterparts. In multiclass spam filtering (Figure 8 in Appendix F), we see that GapLog makes more mistakes than its counterparts for $K > 6$. We suspect this is due to the fact that with logistic loss, the gap map is never zero, which implies that GapLog picks an outcome uniformly at random more often than GapHin and GapSmH, while not gaining any information. Due to this behaviour GapLog makes more mistakes than necessary, which we also observe in the full information setting. In the bandit setting, the additional exploration leads to additional stability for GapLog, as indicated by the small range of performance of GapLog. In all cases, increasing the exploration rate increased the stability of GAPPLETRON, which is very much in agreement with Theorem 5. Additionally, in Appendix F we also compare GAPPLETRON with GAPTRON and in these experiments GAPPLETRON makes less mistakes than GAPTRON.

# 7 Future work

There are several intriguing research directions left open to pursue. While our lower bound holds for general feedback graphs, it is not clear whether our bounds are tight for the bandit setting. Either providing a lower bound or an improved algorithm for the bandit setting remains thus open. Our results show that it is possible to obtain improved bounds for the separable case while maintaining satisfac-

---

[3]This is a version different from the one described by Kakade et al. (2008), in particular, we replaced $\widetilde{U}^t$ in their Algorithm 1 with the gradient of the importance weighted hinge loss.

tory results for the non-separable case. However, as Beygelzimer et al. (2019) show, it is possible to obtain even better guarantees in the separable case of the bandit setting. An algorithm guaranteeing $O(K\|U\|^2)$ mistakes in the separable case and $O(K\sqrt{T})$ surrogate regret in the non-separable case, without prior knowledge of the separability, would therefore be an interesting contribution.

## Acknowledgments and Disclosure of Funding

Nicolò Cesa-Bianchi, Federico Fusco and Dirk van der Hoeven gratefully acknowledge partial support by the MIUR PRIN grant Algorithms, Games, and Digital Markets (ALGADIMAR). Nicolò Cesa-Bianchi was also supported by the EU Horizon 2020 ICT-48 research and innovation action under grant agreement 951847, project ELISE. Federico Fusco was also supported by the ERC Advanced Grant 788893 AMDROMA "Algorithmic and Mechanism Design Research in Online Markets".

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
