# OpenReview forum: "Beyond Bandit Feedback in Online Multiclass Classification"
_NeurIPS.cc/2021/Conference — NeurIPS 2021 Poster_

### Official Review · Reviewer_p3fb · 2021-07-05

**Rating:** 6
**Confidence:** 4

**Summary:**

This paper studies the problem of active online multiclass classification. As opposed to previous works, this paper introduces the additional complexity of gathering data via a feedback graph, as opposed to past works that consider either bandit feedback or the full information case. The authors present an algorithm referred to as Gappletron. The algorithm begins with a max margin prediction of the labels and then breaks into cases. If the action is `revealing’- meaning that its feedback leads to the full information case due to the structure of the feedback graph, then a distribution is computed, and arms appear to be pulled in proportion to an estimated gap. Otherwise, more uniformity enters into the distribution. Following drawing the sample, the algorithm updates loss estimates via IPS. Finally, the algorithm updates a set of weights using the loss estimates and calls out to a generic online convex optimization routine. The feedback graph affects the guarantees that the authors are able to show. The algorithm suffers regret O(B\sqrt(\rho * K * T)) where K is the number of classes and \rho is the domination number of the graph. It should probably be noted that since, as the authors note, finding the smallest dominating subgraph is NP complete, a greedy approximation algorithm must be used and one would instead expect a dependence on \rho * \log(\rho). The authors prove via a lower bound that this complexity is not significantly improvable. They show a lower bound of max(B^2*K , \sqrt(T)). In order to prove their theoretical results, the authors follow the same approach is van der Hoeven 2020 whereby they assume a surrogate loss for the 0/1 loss and control the surrogate regret. The paper concludes with several experiments demonstrating the power of their approach.

**Limitations And Societal Impact:**

Neither are really covered. As the neurips website notes, https://neurips.cc/Conferences/2021/PaperInformation/PaperChecklist, a paper being theoretical is not grounds for the authors ignore the societal impacts of the work. I mention this as a note for future work and not as a contributing factor to my score.



**Main Review:**

Strengths: The setting of a feedback graph is very natural and covers many problems of practical interest. The guarantees in this setting seem natural. The lower bound is a good step towards understanding the complexity of this problem, though they seem to be independent of \rho which is a key parameter for the upper bounds. I also appreciate that the authors provide results both in expectation and with high probability for their regret guarantees. Interestingly, they seem to not be achieved by simply taking \delta = 1/T as has been done in other works. Additionally, the results cover both the separable and non-separable settings, in contrast to prior work. Finally, the algorithm shows strong empirical promise versus the chosen baselines.

Weaknesses: My most major issue is that while the idea is novel, the paper borrows heavily from the techniques of van der Hoeven, 2020- leading to the paper feeling like more of an extension of that work than an independent work. Furthermore, the clarity of the writing could be improved throughout. Especially given the number of assumptions made in this paper, it would have been nice to have a clear, worked out example of all of these assumptions and the guarantees. While I appreciate the level of generality this paper is going for, it would be nice for example to have a quick subsection like one of the subsections of section 4 or 5 in van der Hoeven showing the result in a specific case. Additionally, when the algorithm is presented, smaller theoretical guarantees are interspersed with the description of the functioning of the algorithm, and this is to the detriment of the transparency of both the theory and the algorithm.

Comments:
-	In the intro, the authors note a difference in the order of T in the regret of T^0.5 vs T^2/3. This seems like a major difference, but this is not addressed after the introduction.

Minor comments:
-	Line 131: typo, “equivalent to saying” rather than “equivalent to say”
-	The quantity h(U) is referenced before assignment


**Time Spent Reviewing:**

4.5

---

> ### Author Response · Authors · 2021-08-10
> **Response to reviewer p3fb**
>
> As we acknowledge in the description of Gappletron, the basis of the algorithm relies on exploiting the surrogate gap, an idea introduced by Van der Hoeven (2020). However, we significantly generalize and improve the analysis of this idea. The most obvious generalization is to the feedback graph setting, which reveals the surprising phenomenon that in the worst case, the bandit feedback graph appears to be more difficult than for example the spam filtering feedback graph. A more subtle generalization comes from the design of the gap map. While Van der Hoeven (2020) designs a unique gap map for several surrogate losses, we provide a general gap map under mild regularity conditions on the surrogate loss. Our gap map also allows us to derive Lemma 1, which plays a crucial role in the proof of the high-probability bounds. Another more subtle generalization comes from the more flexible choice of OCO algorithm. Gaptron is restricted to only use OGD with a fixed learning rate, but our analysis reveals that we may actually use an adaptive learning rate. In turn, this adaptive learning rate allows us to provide up to a factor $K$ improvement in the separable case over Gaptron.
>
> As for the description of Gappletron in section 2, we think that Lemma 1, Lemma 2, and the discussion in lines 201-214 are essential to motivate the design and functioning of the algorithm. Indeed, we could move some of the theoretical guarantees to a separate section, but we think that this does the opposite of clarifying the choices that we made while designing the algorithm seeing that the theory and algorithm are intertwined.
>
> As for the number of assumptions made in the paper: other than the regularity assumptions on the surrogate losses all assumptions are standard.
>
> As for the societal impact of our paper, we understand your point. While the paper is mainly theoretical we do agree that theory can and should impact society. Our algorithm can be applied to a vast number of applications such as for example spam filtering or recommendation systems. Even though we introduce a new setting, the goal is similar to other settings where the goal is to provide meaningful guarantees on the number of mistakes a learner makes and therefore there are algorithms in literature with the same goals. While our algorithm may provide significant theoretical and practical improvements over previous algorithms, we do not expect the impact of our research to be substantially different from existing methods.

---

> > ### Comment · Reviewer_p3fb · 2021-08-25
> > **Thank you for your response**
> >
> > Thank you for addressing my questions and clarifying the generalizations of your work over prior art. I am happy to increase my score, though with the extra page, it may be worth adding small discussion comparing these results to the prior art.

---

### Official Review · Reviewer_Qt6c · 2021-07-16

**Rating:** 7
**Confidence:** 4

**Summary:**

This paper considers the problem of online multi-class classification, in which the learner sequentially predicts an output among K classes after observing an input vector. Following the work of van der Hoeven (2020), the authors consider a surrogate regret in which the cumulative number of mistakes incurred by the learner (with zero-one loss) is compared to the cumulative surrogate loss of the best parameter in a ball. The surrogate loss can be the multi-class Hinge loss or the multi-class logistic loss.

This paper generalizes the results and the algorithm of van der Hoeven from bandit and full-information feedback to feedback graphs, proving a new surrogate regret bound that depends on the domination number. Furthermore, they prove the first high-probability regret bound for the problem. Similar to van der Hoeven, they provide improved bound for the separable case and prove a lower-bound.

**Limitations And Societal Impact:**

Yes

**Main Review:**

The paper is generally well written. The main contributions are the following:
- New algorithm (Gappletron) for online multi-class classification. Compared to previous algorithms, the algorithm can deal with general surrogate losses and general feedback graphs, which was not the case for previous analysis.
- New upper-bounds on the surrogate regret. The main novelties are:
    - the bounds hold in high-probability while previous bound were in expectation only;
    - extension of the analysis to feedback graphs: the number of actions from the bandit setting is replaced with the domination number which can be significantly smaller.
- New matching lower-bound that show that sqrt{T} is not improvable with general feedback graphs (which was not the case for full-information). I was a bit disappointed with the fact that the lower-bound only holds for the worst feedback graph. It would have been much stronger if it could depend on the domination number.
- Gappletron seems to significantly outperform the proposed baselines on the synthetic experiments proposed in the paper.
In general, I find the contributions are interesting but not so much surprising. The generalization from bandit feedback to feedback graphs, with bounds that depend on the domination number, is already well studied in the bandit literature. Furthermore, the improvement of the bounds by using the gap between the surrogate loss and the zero-one loss was already present in van der Hoeven (2020). Yet, having a matching lower-bound and having the results in high-probability is an interesting technical contribution.

Comments:
- About the h.p. upper-bound (Thm 2): I unfortunately not had time to go through the proof in details, but I was a bit surprised by its length. Since the bound is of order sqrt{T}, why isn't it possible to just use the bound in expectation together with an Azuma-Hoeffding type inequality (or your Lemma 4) and directly use it together with Thm. 1?
- In your experiments, Gaptron is not used as a baseline in the bandit setting. Does Gappletron in this case coincides with Gaptron?

Typos:
- What is 1_S in Alg 1?
- domain R^{Kxd} as input in Alg 1
- Initialize W_t in Alg 1
- What is delta' in Thm 5?
- I would define hat g_t with hat ell_t in line 208. And then say that for the full info setting \hat \ell_t = \ell_t

**Time Spent Reviewing:**

3

---

> ### Author Response · Authors · 2021-08-10
> **Response to reviewer Qt6c**
>
> To address your comment on the high-probability bound, consider the following. We need to control the concentration of the estimated surrogate losses around their means. In order to do so we may employ Lemma 4, which gives us equation (14) in the appendix. The $V_{\max}$ term in that equation is $O(\sqrt{T})$ which means equation (14) is $O(T^{3/4})$. So we need to further use the negative terms due to the particular predictions of Gappletron to control equation (14). A similar analysis happens in the full information setting, which we sketch below Theorem 2. If we would have stopped the analysis after the displayed equation in line 277, we would obtain a $O(K h(U)^2 + \sqrt{T})$ upper bound. Instead, we use the negative terms to prove a $O(K h(U)^2)$ upper bound.
>
> In the experiments, Gaptron would not coincide with Gappletron. Gaptron uses a different gap map and a different definition of the logistic loss. Gaptron and Gappletron both use OGD, but the analysis of Gappletron reveals that we may use an adaptive learning rate for OGD as opposed to the worst case learning rate. The exploration rate is also different. However, Gaptron was overlooked when designing the experiments and we will add it to the comparison.
>
> Thank you for pointing out the typos, which we will address.

---

> > ### Comment · Reviewer_Qt6c · 2021-08-19
> > **Quick response to the authors**
> >
> > I thank the authors for their response to my review. While I am currently travelling (for the next 1.5 week) and in a very tight time schedule,  I only give a high level response. I also read other reviews.
> >
> > I acknowledge that a lower-bound in sqrt{T} is already an interesting contribution that point a nice phenomenon: in the worst case, the bandit feedback graph appears to be most difficult one. I thank the authors for their details about the high-probability bound which was indeed not trivial. I am happy to increase my score to 7.

---

### Official Review · Reviewer_TAuC · 2021-07-16

**Rating:** 7
**Confidence:** 1

**Summary:**

The authors study the problem of multiclass classification where the feedback is determined by an arbitrary directed graph, where each node corresponds to an action by the learner and has at least one incoming edge, and if there is an edge from node a to b, then the learner, on performing action a, observes the loss of action b. This model is a generalization of the bandit feedback case, and has applications in other problems. They present GAPPLETRON, an online multiclass algorithm for arbitrary feedback graphs, and prove surrogate regret bounds. They also present experimental results that show competitive performance of GAPPLETRON against various baselines.


**Limitations And Societal Impact:**

While not limitations, the authors do discuss potential improvements to the result, and have mentioned that since their result is mainly theoretical there is no foreseeable societal impact.

**Main Review:**

The problem is well motivated and the results seem novel. While the result is well presented and appears to be correct, I am unable to accurately review the paper in the context of originality and significance because I feel like I lack the domain knowledge to accurately assess the paper.

**Time Spent Reviewing:**

4-5

---

### Official Review · Reviewer_u4LL · 2021-07-17

**Rating:** 6
**Confidence:** 2

**Summary:**

This study deals with (adversarial) online multiclass classification problem with partial feedback about the true label.
The feedback is characterized by a directed graph on the label space,
through which the model includes the full-information setting, the bandit feedback setting, the spam-filtering problem setting,
and so on.
For this general problem setting,
an algorithm called GAPPLETRON is developed.
The performance is measured in terms of the surrogate regret,
the difference between the number of mistake and cumulative loss for an optimal model.
The proposed GAPPLETRON algorithm achieves nearly optimal surrogate-regret bounds both for non-separable and for separable cases.

**Limitations And Societal Impact:**

A more detailed explanation of the limitations resulting from the regularity assumption (Eq. (2), (3)) would be appreciated.
For example, it would be good to have information on which existing studies meet this assumption and which do not.

**Main Review:**

Originality:

To the best of my knowledge,
this is the first study that deals with the online multiclass classification with feedback graph.
However,
as mentioned in the paper,
each of online multiclass classification and the model of feedback graph is not new.
Since it is a natural idea to combine these elements, the problem setting itself is not very original.
The proposed algorithm is based on unbiased estimator for loss functions and online convex optimization,
which is a standard approach for bandit optimization problems.
The surrogate-regret analysis for the proposed algorithm appears to contain new ideas,
which relay on the regularity of the surrogate losses.

Quality:

As far as I can see, no technical errors have been found.
I think the authors are careful and honest about evaluating both the strengths and weaknesses of their work.

Clarity:

The main part of the paper is clearly written.
One question I had was whether the proposed algorithm could be compared with BANDITRON by Kakade et al.(2008),
which has been analyzed for the hinge loss (with $\kappa = 0$).
In my understandint,
as the hinge loss does not suffice the assumptions of (2) and (3),
Gappletron has not yet been proven to be superior to BANDITRON.
In my understanding is correct,
the description in lines 84--98 may be misleading.
I think it is important to emphasize that BANDITRON employs hinge loss and that the analysis for Gappletron does not directly apply.

Significance:

This work considers a new problem setting, which will be of interest to the research community of online learning.
The proposed algorithm is simple and works well for various situations including non-separable cases.
I think that this work has a certain impact on the research community online learning.
It would be better if
the dependence of regret bounds on graph-structure parameters (e.g., the parameter $\rho$)
could be analyzed more precisely,
as in the literature by Alon et al.(2015) and by Alon et al.(2017).
I'm also interested in cases where the regularity assumptions are relaxed.

**Time Spent Reviewing:**

5

---

> ### Author Response · Authors · 2021-08-10
> **Response to reviewer u4LL**
>
> Your comment regarding the comparison with Banditron is correct. While for the separable case the comparison is fair as both surrogate loss functions become 0, in the non-separable case Gappletron uses a surrogate loss function different from the one used in Banditron. We will make this point clearer in the revision.
>
> As for the limitations resulting from the regularity assumption, the only surrogate loss we found in the literature that does not satisfy the regularity assumptions is the hinge loss with $\kappa = 0$. We will provide some intuition as to why this is the case for $K = 2$. For $K = 2$ a surrogate loss is usually a function of the margin $y W^\top x$, where $y \in [-1, 1]$. This means that assumption (3) reduces to a smoothness condition on a one-dimensional function and a boundedness condition on the feature vectors. Assuming $\ell(W, x, y) = f(y W^\top x)$ with convex $f$, assumption (2) can be easily verified via an application of Jensen's inequality. For $K > 2$, surrogate losses are often designed using a generalization of the margin, such as for example in line 461, for which a similar approach can be applied to prove the regularity assumptions.

---

### Author Response · Authors · 2021-08-10
**General response**

We thank the reviewers and area chairs for their careful reading of our paper.

While we agree with reviewers u4LL, Qt6C, and p3fb that a lower bound which furthers our understanding the dependence on $\rho$ would be interesting, we would like to emphasize that our lower bound does provide a new understanding of the dependence on $T$.

Indeed, the generalization from bandit feedback to graph feedback has been well studied in the bandit literature. However, as commented by reviewer p3fb, Alon et. al. (2017) provide a $\Omega(T^{2/3})$ lower bound for worst-case feedback graphs, while a $\Omega(\sqrt{T})$ lower bound is known in standard bandit literature. The worst-case feedback graph of Alon et. al. (2017) has $\rho = 1$, which in our setting would correspond to, for example, spam filtering. That is precisely the case where our lower bound matches our upper bound. Interestingly, the case where our lower bound is not tight is for the bandit setting, which for Gappletron is the worst-case feedback graph! This is a remarkable contrast between the setting considered in the literature and our setting, and we consider this a surprising and interesting phenomenon revealed by our analysis.

---

### Decision · Program_Chairs · 2021-09-28

**Decision:**

Accept (Poster)

**Comment:**

This paper studies a new problem setting of online multiclass linear classification with graph-structured feedback. The reviewers agree that it is of interest to the online   learning research community. In summary, the paper:
- proposes the Gappletron algorithm that can establish regret guarantees against a wide range of surrogate loss functions;
- proves high probability regret bounds that are new
- gives a new \sqrt{T} regret lower bound
- gives improved experimental results against prior art

The reviewers also pointed out that studying: (1) the regret's fundamental dependence on the graph independence number \rho (e.g. by showing lower bounds in term of \rho); (2) whether the results can be extended to the commonly-used hinge losses, are important future directions. The reviewers would also like the authors to incorporate their rebuttal on comparison with prior art into the final version.

**Consistency Experiment:**

NeurIPS has a long history of experimentation. In 2014, NeurIPS ran an experiment in which 10% of submissions were reviewed by two independent committees to quantify the randomness in the review process. This year, we repeated a variant of this experiment to see how the quality of the review process has changed over time.  This paper was part of the experiment and was therefore assigned to two committees (consisting of reviewers, an Area Chair, and a Senior Area Chair) that reached independent decisions.  If both committees made the same recommendation, this recommendation was followed. If a single committee recommended acceptance, the paper was accepted (with the exception of a few cases in which the other committee identified what we considered a fatal flaw, e.g., an error in a key result).

This copy’s committee reached the following decision: **Accept (Poster)**

The other committee assigned to the paper recommended **Reject**.  You can find the other set of reviews, along with any follow up discussion with the authors here:
https://openreview.net/forum?id=cZpUtLSOJnu